# Encouraging Split Liver Transplantation for Two Adult Recipients to Mitigate the High Incidence of Wait-List Mortality in the Setting of Extreme Shortage of Deceased Donors

**DOI:** 10.3390/jcm8122095

**Published:** 2019-12-01

**Authors:** Kun-Ming Chan, Yu-Chao Wang, Tsung-Han Wu, Chih-Hsien Cheng, Chen-Fang Lee, Ting-Jung Wu, Hong-Shiue Chou, Wei-Chen Lee

**Affiliations:** Department of General Surgery and Chang Gung Transplantation Institute, Chang Gung Memorial Hospital at Linkou, Chang Gung University College of Medicine, Taoyuan 33302, Taiwan

**Keywords:** split liver transplantation, adult recipient, graft weight, risk factor, outcome

## Abstract

Background: Organ demand for liver transplantation (LT) is constantly increasing. Split liver transplantation (SPLT) is an ideal option for increasing the number of available liver grafts for transplantation and ameliorating organ shortage to a certain degree. However, SPLT for two adult recipients is still not broadly applied. Methods: We retrospectively analyzed the outcomes of SPLT for adult recipients at a single center. All donor, recipient, and transplantation factors were thoroughly investigated to clarify factors affecting patient outcomes after LT. Results: One hundred consecutive adult SPLTs were performed during the study period. Early mortality and 1-year mortality occurred in 21 and 31 recipients, respectively. On multivariate analysis, graft weight (*p* = 0.036, odds ratio = 0.99, 95% confidence interval = 0.98–0.99) was the independent risk factor associated with early mortality; however, no factor was significantly related to 1-year mortality. On receiver operating characteristic curve analysis, a graft weight of 580 g was identified the cutoff for stratifying outcomes. Recipients transplanted with a graft weighing ≥580 g had significantly better outcome as compared with other recipients (*p* = 0.001). Moreover, SPLT remarkably provided a better survival benefit for recipients than those on the LT wait-list (*p* < 0.0001). Conclusions: Given the considerable incidence of wait-list mortality, SPLT for two adult recipients should be encouraged whenever possible to increase the donor pool and benefit patients awaiting LT. Nonetheless, caution should be taken with a smaller graft weight owing to the risk of early graft loss.

## 1. Introduction

Since the first liver transplantation (LT) a half century ago, LT has evolved into a common operation in many transplantation centers nowadays. [1,2] With the growing experience, this highly complex operation has become more feasible and has achieved satisfactory outcomes owing to advances in surgical techniques and perioperative patient care. However, there remain several unmet needs for improving the overall perspective of LT. Most important, there is a persisting shortage in the availability of organ donations for LT. The gap between organ donation and the demand for transplantation is constantly widening, and the scarcity of liver grafts is one of the main obstacles to LT. Therefore, living donors have become an alternative source of liver grafts for transplantation in order to overcome this problem. Although living donor LT (LDLT) has been increasingly performed, efforts to maximize the utilization of organs from deceased donors are still needed. Accordingly, split LT (SPLT) has been introduced as an ideal option for enabling LT in which one donor could benefit two recipients, and this method might also help ameliorate the organ shortage to a certain degree.

SPLT was initially proposed as a method in which an adult liver graft is divided into a smaller graft for a pediatric recipient and a bigger graft for an adult recipient [3]. To date, this remains the main approach in SPLT by which excellent outcomes can be expected [4,5,6]. Meanwhile, SPLT in which one liver is divided into two full (left hemi-liver and right hemi-liver) grafts for two adult recipients has also been performed [7,8]. However, splitting the liver during graft procurement is technically more challenging than transplanting the whole liver. In addition, the many potential complications after SPLT remain a matter of concern [9,10,11]. Therefore, SPLT for two adult recipients is still not common practice.

Nonetheless, deceased organ donations for LT are extremely scarce in Asian countries, including Taiwan [12]. Thereby, SPLT may be a promising strategy to expand the donor pool in order to mitigate organ shortage and provide an additional opportunity for LT in adult patients. As previously reported from the institute, the initial results showed that SPLT for two adult recipients was feasible and comparable with those of LDLT in the current Model for End-Stage Liver Disease (MELD) era [13]. Therefore, in this study, we gathered more experiences and retrospectively analyzed patients who underwent SPLT at a single center in the setting of scarce organ donations. Additionally, prognostic factors associated with patient outcomes and the beneficial effects of SPLT in patients were examined through all donor, recipient, and transplantation factors. This study also aimed to provide additional information for guiding decision-making and optimizing strategies in SPLT for adult patients.

## 2. Materials and Methods

### 2.1. Recipient Cohort

Among 1093 LTs, a total of 100 consecutive adult SPLTs were performed between July 2003 and June 2019 at our transplantation institute. All medical records of donors and recipients were retrospectively reviewed, and data were collected for analysis with approval from the Institutional Review Board of Chang Gung Memorial Hospital (approval no. 98-3794B). Owing to the retrospective nature of the study, the need for informed consent from patients was waived. The study protocol complied with the ethical standards established by the Declaration of Helsinki with respect to confidentiality of patient data. Importantly, no organs from executed prisoners were used in this study. Further, patients registered in the wait-list for LT and patients transplanted with a whole liver graft during the corresponding period were analyzed for survival comparison.

### 2.2. Donor Evaluation

Generally, the potential donor should be hemodynamically stable with acceptable cardiopulmonary function, no uncontrollable bacterial infection, and no malignant neoplasm. All potential donors were thoroughly evaluated for eligibility for liver donation with biochemical tests and hepatic ultrasonography. The donor should also have acceptable liver functional reserve and only mild fatty change in the hepatic parenchyma.

The decision to split liver grafts was based on the evaluation of the donor, evaluation of the liver graft, and recipient allocation from the national organ sharing program, as previously described [13,14]. Briefly, the donor should have stable hemodynamics, well-preserved liver function, only mild fatty change in the liver parenchyma, and age between 15 and 55 years. The expected liver volume of split hemi-liver grafts was estimated by calculating the standard liver volume (SLV) [15] and using the equation that includes the maximum diameter of the portal vein, as described by our institute after 2011 [14]. According to the allocation system of the national organ sharing program, the first hemi-liver graft was allocated to the first priority recipient in the wait-list with the highest MELD score. The other hemi-liver graft was allocated to a size-matched recipient in the wait-list. Generally, the first priority recipient should have an estimated graft–recipient weight ratio (GRWR) of no less than 0.8%. Otherwise, the potential liver graft would be transplanted as a whole liver.

### 2.3. Liver Graft Preparation

Separation of the hepatic parenchyma for split liver grafts was generally performed in situ, as previously described, except for one case in which hemodynamic instability occurred in the donor during graft procurement [13]. During graft recovery, the liver was first inspected to evaluate the sizes of the right and left lobes, the anatomy of the hepatic artery, and the consistency of the hepatic parenchyma. Intraoperative cholangiography was routinely performed in all donors to determine the cut point of the left and right hepatic ducts. The venous tributaries of the middle hepatic vein (MHV) were assessed using intraoperative ultrasonography to demarcate the transection line that is usually along Cantlie’s line, and the main trunk of the MHV and the caudate lobe were kept in the left hemi-liver graft.

All hepatic parenchyma transections were performed using the Cavitron Ultrasonic Surgical Aspirator (Valleylab Inc., Boulder, CO, USA), and the Pringle maneuver for inflow vascular control was not applied during hepatectomy. All vessels between the right and left lobes were divided, and the main tributary veins of segments 5 and 8 in the right lobe were temporarily clamped with bulldog clamps and released when graft perfusion was commenced. The hepatic parenchyma was split down to the anterior surface of the inferior vena cava (IVC) when possible, and the hepatic duct was divided. Subsequently, the liver graft was perfused with histidine-tryptophan-ketoglutarate (Essential Pharma, Newtown, PA, USA) solution and allowed to recover.

The right and left hemi-liver grafts were completely separated on the back table, and all vascular structures were divided (Figure 1A). Generally, the common hepatic artery was kept in the left hemi-liver graft and the IVC was preserved in the right hemi-liver graft. Meanwhile, the main tributary veins of segments 5 and 8 were reconstructed with a venous graft and also drained into the IVC of the graft. The venous drainage of the caudate lobe was reconstructed whenever size ≥3 mm to preserve graft volume and function, especially when GRWR was borderline as previously described from the center [16].

### 2.4. Recipients and Liver Graft Implantation

Generally, all potential recipients were routinely evaluated for laboratory tests of infectious microorganisms, color Doppler cardiac sonography, whole-body computed tomography (CT), and panendoscopy prior to list on the wait-list or transplantation. An informed consent document that detailed the indication and risk of LT were well explained to the recipient and were to be signed before transplantation. All recipient operations started with total hepatectomy with IVC preservation. The right hemi-liver graft was orthotopically implanted using the piggyback technique (Figure 1B). The left hemi-liver graft was either orthotopically implanted at the left abdominal cavity (Figure 1C) or heterotopically implanted into the right subphrenic space which is a method previously described by our institute and termed as “left at right LT” [17,18]. Subsequently, the liver graft was reperfused after portal vein reconstruction. The hepatic artery was reconstructed using microscopic vascular anastomosis or the Carrel patch technique depending on the size of the artery of the graft. Finally, the bile duct was reconstructed through end-to-end anastomosis without T-tube stenting.

### 2.5. Postoperative Care and Statistical Analysis

After transplantation, all recipients were transferred to the intensive care unit for postoperative management. Biochemical tests and hepatic ultrasonography were performed at regular intervals. The postoperative immunosuppressive regimen consisted of a combination of methylprednisolone, tacrolimus, and mycophenolate mofetil and was adjusted according to the clinical status of the recipient.

The endpoint outcome measures were early mortality within 3 months, 1-year mortality, and overall survival (OS). The OS was calculated from the date of LT to the date of death or the end of the study, and all causes of deaths were counted as events. Survival curves were generated using the Kaplan–Meier method and compared using the log-rank test. Continuous variables were compared using the Mann–Whitney U test, and categorical variables were compared using the χ^2^ test or Fisher’s exact test, as appropriate. In order to include all important factors, variables with a *p*-value < 0.10 from univariate analysis were selected for multivariate analyses by enter procedure using logistic regression models. All statistical analyses were performed using the statistical software SPSS version 20.0 (SPSS Inc., Chicago, IL, USA) for Windows. A *p*-value of <0.05 was defined as statistically significant.

## 3. Results

### 3.1. Clinical Features of Donors

In accordance with the organ donation regulation in Taiwan, all livers were procured after the donors were declared brain dead by specialists. None of the liver grafts in this study were procured from a donor with a non-beating heart. Table 1 summarizes the characteristics of all donors. A total of 108 hemi-liver grafts were procured from 54 donors, in which 9 donors were from the national organ sharing program that provided 14 hemi-liver grafts, and 45 donors from the institute that provided 86 hemi-liver grafts for SPLT in the center. The remaining 8 hemi-liver grafts were shared with other transplantation centers for LT. These donors consisted of 39 men and 15 women, with a median age of 27 years (range, 15–53 years). The major causes of brain death were cerebrovascular accidents in 29 donors (53.7%) and traumatic head injury in 17 donors (31.5%). The median hospitalization duration was 5 days (range, 1–36 days). There were 9 donors who had undergone cardiopulmonary resuscitation at hospital arrival and whose hemodynamics became stable thereafter. The median estimated SLV was 1274 mL (range, 929–1533 mL), and the median actual liver weight was 1380 g (range, 990–2100 g). The discrepancy between the actual liver weight and the estimated SLV ranged from –32.8% to +37.1%, with a median of –4.5%.

### 3.2. Outcome Analysis of Recipients

After SPLT, the median follow-up period was 17.2 months (range, 1 day to 196.4 months). During the follow-up period, 54 (54%) recipients were still alive by the end of this study. Early hospital mortality (from 1 to 86 days after LT) occurred in 21 (21%) recipients. Ten recipients died during the period from 3.6 to 11.8 months after LT. The remaining 15 deaths occurred 1 year after SPLT.

Table 2 summarizes the outcomes in terms of early mortality, in which all factors with a *p*-value < 0.1 in univariate group comparison analysis, including graft type (*p* = 0.004), graft weight (*p* = 0.004), and GRWR (*p* = 0.054), were selected into multivariate analysis. The results of multivariate analysis showed that graft weight (*p* = 0.036, odds ratio = 0.99, 95% confidence interval [CI] = 0.98–0.99) was the only independent factor for the occurrence of early mortality. 

Outcomes related to 1-year mortality were further analyzed. In this analysis, 11 recipients who did not complete the 1-year follow-up period were excluded. The results showed that four factors (*p* < 0.1)—graft type (*p* < 0.001), graft weight (*p* < 0.001), cold ischemia time (*p* = 0.039), and GRWR (*p* = 0.007)—were strongly related to 1-year mortality. Furthermore, multivariate regression analysis of these four factors revealed that only two factors—graft weight (*p* = 0.070) and cold ischemia time (*p* = 0.069)—showed borderline correlation with 1-year mortality (Table 3).

On the basis of the results of the analysis of predictors, graft weight was evaluated using the receiver operating characteristic (ROC) curve to determine the best cutoff point related to the outcome of SPLT. A graft weight of 580 g was identified as the most optimal cutoff point for stratifying outcomes. The area under the ROC curve was 0.71 (95% CI = 0.59–0.82) for early mortality. The sensitivity and 1-specificity were 71.4% and 30.3%, respectively. Furthermore, recipients who were transplanted with grafts weighing <580 g and encountered early mortality were analyzed. The clinical features and major events leading to early mortality for these patients are summarized in Table 4. Of these, 7 recipients encountered graft dysfunction that was characterized by serum bilirubin ≥10 mg/mL on postoperative day 7 and prolonged cholestasis [19] and followed by complicated bacterial infections. Two recipients (No. 342, 426) had antibody mediated rejection with a clinical course as previously described [20]. One recipient (No. 126) had acute renal failure followed by hyperkalemia and unstable arrhythmia leading to death at 9 days after LT, and one recipient (No. 848) had HCV relapse followed by graft failure. Two recipients (No. 399, 1033) were mortalities because of pneumonia and intracranial hemorrhage after LT, respectively. One recipient (No. 790) was initially recovering very well with a normal graft function but subsequently developed acute graft-versus-host disease [21], and one recipient (No. 978) died at the time of transplantation because of massive bleeding.

### 3.3. Survival Analysis

The 1-, 3-, and 5-year survival rates in all recipients after SPLT were 68.3%, 60.5%, and 59.0%, respectively. In the corresponding period, 165 adult patients (132 men and 33 women) who had undergone LT with a whole liver graft and 995 adult patients (750 men and 245 women) who were registered in the wait-list for LT and had not received any type of LT were analyzed for survival comparison. Of those, 816 patients (82%) had no opportunity to undergo LT and died during the waiting period (range, 1 day to 115 months), among whom 593 patients (59.6%) died within 1 year of being in the wait-list. The remaining 179 patients were still awaiting LT (range of waiting period, 0.4–136 months) by the end of this study. The 1-, 3-, and 5-year survival rates of patients in the wait-list were 40.2%, 16.3%, and 8.0%, respectively. The comparison of survival curves showed that SPLT remarkably provided a survival benefit for patients who had been indicated for LT and registered in the wait-list (Figure 2, *p* < 0.0001). However, the survival curve of SPLT was comparable with whole liver transplantation, in which the 1-, 3-, and 5-year survival rates for whole liver recipients were 73.4%, 66.9%, and 66.9%. (Figure 2, *p* = 0.198) The clinical features of patients who had listed in the wait-list or underwent LT is summarized in Table 5. Significantly, patients who had undergone either SPLT or whole liver transplantation had a relative higher MELD score than that of patients in the wait-list.

The comparison of survival curves in patients after SPLT based on graft weight is illustrated in Figure 3. As expected, recipients with a graft weight of <580 g had significantly inferior survival than recipients transplanted with a graft weight of ≥580 g. The OS at 1, 3, and 5 years was 82.7%, 72.1%, and 69.6%, respectively, in the larger graft group (≥580 g), whereas the corresponding OS was 45.1%, 42.1%, and 42.1%, respectively, in the smaller graft group (<580 g) (*p* = 0.001).

## 4. Discussion

This study describes the experience of 100 SPLTs performed in adult recipients at a single center in the setting of organ donor shortage. With the achievement of this milestone after a long evolution of the procedure, we aimed to thoroughly assess the beneficial effects and risk factors of SPLT in order to provide additional information for optimizing the strategies in this approach. Importantly, the results showed that graft weight was strongly associated with the initial success of SPLT. Further, SPLT for two adult recipients saved patients from the long waiting period for LT, which is characterized by a high mortality rate.

It is well known that extreme organ shortage is the most important hindrance to organ transplantation worldwide. Specifically, deceased organ donors are extremely scarce in the East Asian region. As a result, LDLT is increasingly performed nowadays to overcome the shortage of liver donations for transplantation [22]. However, splitting the liver into two hemi-liver grafts might be another promising strategy to maximally utilize an organ from a deceased donor as well to increase the donor pool for LT. Currently, the main approach in SPLT remains to divide a whole liver into a smaller graft for a pediatric recipient and a bigger graft for an adult recipient. Although splitting a whole liver into a full right hemi-liver graft and a full left hemi-liver graft for two adult recipients has shown promising results, this method is still relatively less performed than the aforementioned approach [23].

Historically, a number of reasons may be responsible for the less frequent use of SPLT for two adult recipients. First, splitting the liver into two hemi-liver grafts during donor operation is technically demanding [24,25]. Second, dividing a good quality graft into two marginal hemi-liver grafts might simultaneously endanger the two recipients [26,27]. Third, the allocation of two split liver grafts could be a concern when the recipients are in two different centers [28]. Fourth, a considerable number of complications have been reported to be associated with dividing and sharing the vascular and biliary structures between full right/left hemi-liver grafts [29,30]. However, owing to the growing experience as well as the evolutions in the surgical technique and perioperative patient care for LT nowadays, these obstacles are no longer insurmountable. Although most of the previous studies included a small number of patients, the overall number of SPLTs for two adult recipients is increasing [23,31].

Additionally, the allocation system and accessibility of the transplantation facilities are important considerations with respect to the practicability of SPLT. Generally, SPLT for a pediatric recipient and an adult recipient is better accepted by most transplantation programs when the split liver grafts are allocated to two different centers. On the other hand, most transplantation centers are reluctant to use leftover hemi-liver grafts gained from SPLT for two adult recipients. In such circumstances, trust and collaboration between different centers are essential for successfully sharing hemi-liver grafts. Currently, there is no formal allocation system, particularly for sharing hemi-liver grafts from SPLT between transplantation centers in Taiwan. As a result, 46 pairs of SPLTs were performed at our institute and only 8 hemi-liver grafts were shared with other transplantation centers.

However, the accessibility of the facilities in terms of space and manpower might provide a large challenge to performing SPLT for two recipients at the same time. Accordingly, three teams need to be activated for simultaneous organ procurement and recipient operations. There were only four transplantation surgeons at our institute by the year 2008; however, the number has doubled to date. Although the outcomes of SPLT were obviously not affected by the different eras before and after 2008 in this study, the greater number of transplantation team members largely provided relief to surgeons from the stress of the transplantation operations. Moreover, sleep deprivation and fatigue have been concerns with respect to physician performance and patient outcomes. [32,33]. Thus, this study also compared the time of transplantation and the patient outcomes, and no significance was observed between daytime and nighttime transplantation in terms of recipient mortality. This reflects the consistent performance of LT and the feasibility of practicing SPLT in the transplantation center during the study period.

Nonetheless, although the surgical technique may be the most important factor for the accomplishment of LT, there remain several concerns affecting the long-term graft and patient survival. Generally, determining the feasibility of LT usually starts from donor and recipient matching, and GRWR has been a common indicator for assessing the suitability of a partial liver graft for transplantation. Although a minimum GRWR of 0.6% could be successfully utilized in LDLT, a GRWR ranging from at least 0.8% to 1.0% is recommended for split liver grafts from deceased donors [13,24,34]. However, GRWR is probably not the only factor related to patient outcome in SPLT, as graft failures have also been reported in cases of both high and low GRWRs [35,36]. In line with those reports, GRWR was not a significant factor associated with graft loss in SPLT for two adult recipients in this study.

On the other hand, graft weight was a significant factor leading to early mortality in adult SPLT in this study. Although many equations were utilized for calculating the SLV, none of them were able to estimate the true liver volume and weight. Generally, liver volume is not equal to liver weight because many factors could affect the specific gravity of the liver. This discrepancy was observed not only in this study but also in other previous studies [37,38]. To our knowledge, no efficient method could precisely predict the liver graft weight before liver recovery. Therefore, hemi-liver grafts from the split liver of a deceased donor may not match the intended adult recipients.

The limitation of the study might be its retrospective nature with a small number of patients. Although SPLT for two adult recipients was performed whenever possible, the number of patients remain small because of the scarcity of deceased donors during the long study period. Additionally, this study might be considered impractical in the clinical setting despite establishing a cutoff graft weight (580 g). Meanwhile, the odds ratio of a smaller graft for early mortality is only 1% to 2% difference in the study. Although the study cautions that a graft weight less than the cutoff value may pose a risk of early graft loss, SPLT should not be discouraged under this concern. Moreover, the small-for-size problem might be overcome in the near future owing to the growing knowledge and experience of small grafts. Although the outcome of SPLT was comparable with that of whole liver transplantation in this study, the overall outcome might be criticized for the inferior survival as compared with the outcomes of other reports from whole liver transplantation. However, the important observations might provide additional information for guiding decision-making in the selection of a suitable deceased donor for SPLT to two adult recipients.

In summary, LT has evolved into a common operation with satisfactory outcomes nowadays. However, SPLT for two adult recipients is still not a popular practice worldwide. Given the considerably high incidence of wait-list mortality, SPLT for two adult recipients should be encouraged to increase the donor pool and benefit patients awaiting LT.

## Figures and Tables

**Figure 1 jcm-08-02095-f001:**
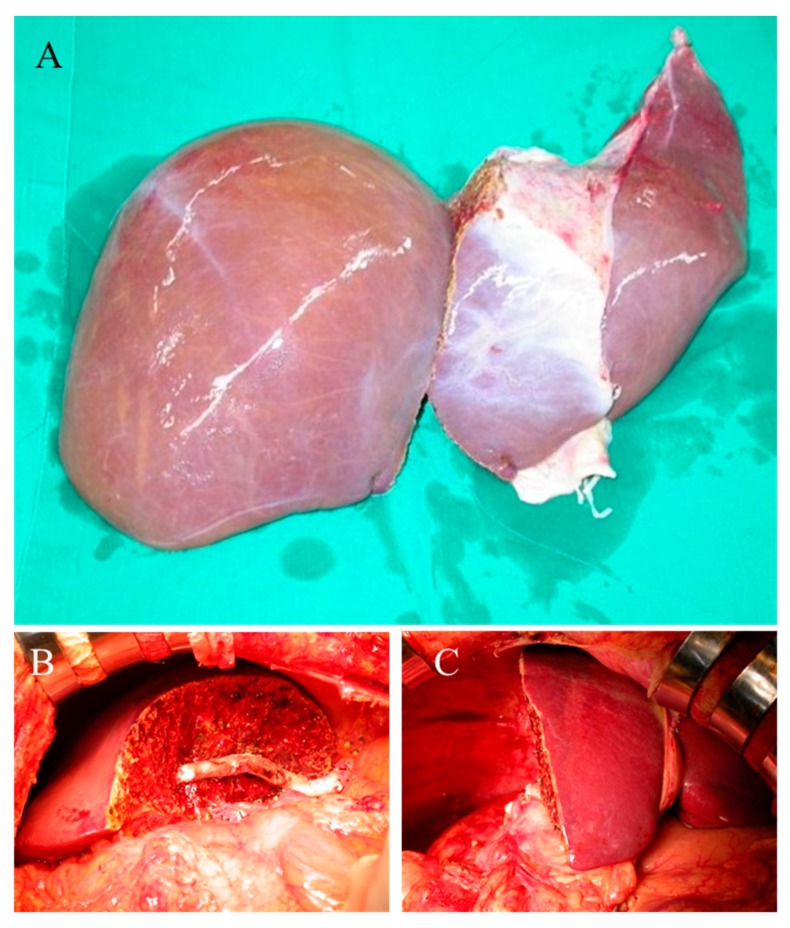
Illustration of split liver transplantation. (**A**) The right and left hemi-liver grafts were completely separated on the back table. (**B**) Right hemi-liver graft implantation. (**C**) Left hemi-liver graft implantation.

**Figure 2 jcm-08-02095-f002:**
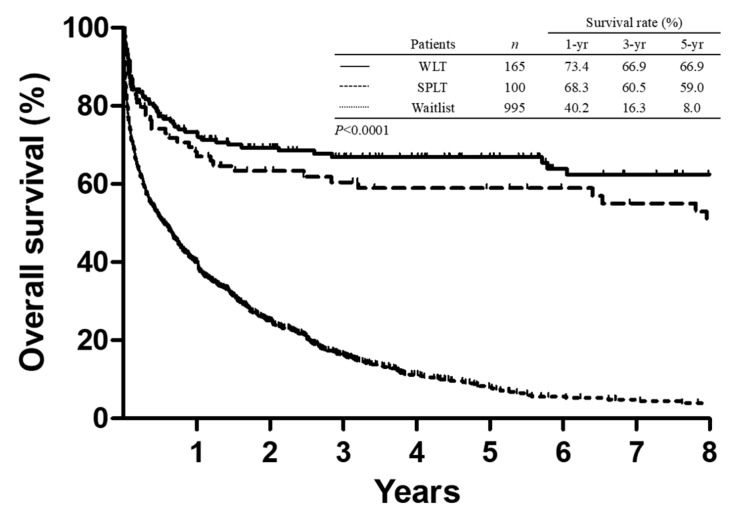
Comparison of Kaplan–Meier survival curves between patients who underwent liver transplantation (LT) and patients in the wait-list. Split liver transplantation (SPLT) remarkably provided a survival benefit for patients (*p* < 0.0001). No significant difference between whole liver transplantation (WLT) and SPLT (*p* = 0.198).

**Figure 3 jcm-08-02095-f003:**
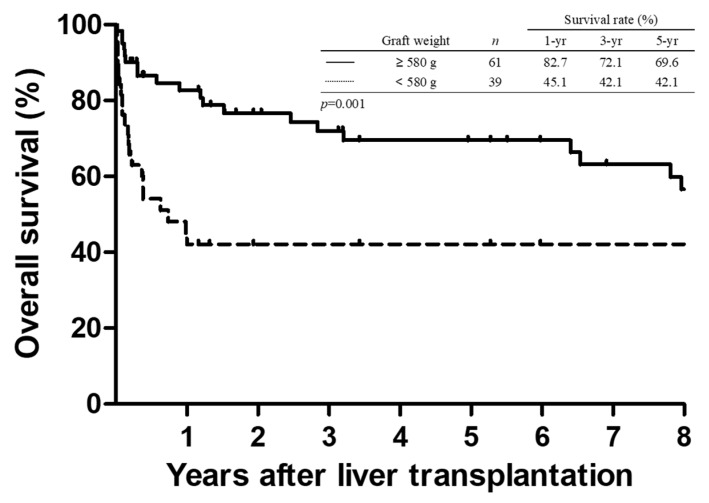
Comparison of Kaplan–Meier survival curves in patients who underwent split liver transplantation based on the graft weight. Recipients transplanted with a graft weight of ≥580 g had significantly superior survival than the other recipients (*p* = 0.001).

**Table 1 jcm-08-02095-t001:** The clinical characteristics of deceased donors.

Total Donors	*n* = 54
Male:female	39:15
Age (years)	27 (15–53)
Body height (cm)	170 (150–187)
Body weight (kg)	67 (45–97)
Estimated standard liver volume (mL)	1274 (929–1533)
Actual liver graft weight (gm) *	1380 (990–2100)
Discrepancy of graft estimation (%)	–4.5% (–32.8%–37.1%)
Hospital stay (days)	5 (1–36)
CPCR history	9
Cause of brain death	
CVA	29
Traumatic head injury	17
Others	8
Graft recovery	
In the institute (*n* = 45)	
Two hemi-live graft	41 (82 hemi-liver graft)
One hemi-liver graft	4 hemi-liver graft
Other institute (*n* = 9)	
Two hemi-liver graft	5 (10 hemi-liver graft)
One hemi-liver graft	4 hemi-liver graft

* Actual liver weights were the sum of left and right liver graft weight; CPCR, Cardiopulmonary Cerebral Resuscitation; CVA, cerebrovascular accident.

**Table 2 jcm-08-02095-t002:** Analysis of risk factor for early mortality.

	Univariate Analysis			Multivariate Analysis	
Characteristics	Yes, *n* = 21	No, *n* = 79	*p*-Value	OR (95% CI)	*p*-Value
Donor factors					
Age, median (range)	32 (15–49)	27 (15–53)	0.611	—	
Sex (Male:Female)	13:8	62:17	0.119	—	
Cause of brain death					
CVA	16 (76.2)	58 (73.4)	0.848	—	
Traumatic head injury	4 (19.0)	13 (16.5)			
Others	1 (4.8)	8 (10.1)			
Hospital stay (days)	5 (1–36)	5 (1–36)	0.835	—	
Graft factors					
Graft type			0.004	1.30 (0.31–5.38)	0.710
Left hemi-liver	16 (76.2)	32 (40.5)			
Right hemi-liver	5 (23.8)	47 (59.5)			
Graft weight (gm)	520 (335–1100)	695 (350–1500)	0.004	0.99 (0.98–0.99)	0.036
Cold ischemia time (min)	474 (102–763)	337 (69–779)	0.207	—	
Warm ischemia time (min)	39 (31–57)	45 (27–65)	0.116	—	
Recipient factors					
Age, median (range)	49 (39–65)	51 (33–65)	0.929	—	
Sex (Male:Female)	12:9	51:28	0.532	—	
Main indication of transplantation			0.353	—	
Alcoholic Liver cirrhosis	3 (14.3)	20 (25.3)			
Virus-related liver cirrhosis	7 (33.3)	24 (30.4)			
Hepatocellular carcinoma	5 (23.8)	24 (30.4)			
Others	6 (28.6)	11 (13.9)			
Viral hepatitis			0.902	—	
HBV	10 (47.6)	35 (44.3)			
HCV	4 (19.0)	15 (19.0)			
HBV + HCV	0	2 (2.5)			
None	7 (33.3)	27 (34.2)			
Child classification			0.220	—	
A, B	13 (61.9)	37 (46.8)			
C	8 (38.1)	42 (53.2)			
Chronic kidney disease staging			0.462	—	
Grade 1,2	14 (66.7)	59 (74.7)			
Grade 3,4,5	7 (33.3)	20 (25.3)			
Hemodialysis before transplantation	3 (14.3)	5 (6.3)	0.232	—	
MELD score, median (range)	18 (7–42)	19 (7–45)	0.842	—	
Biochemical factors					
Sodium (Na, ng/mL)	137 (121–151)	138 (124–152)	0.176	—	
Total Bilirubin (ng/mL)	3.7 (0.5–34.3)	4.1 (0.5–38.2)	0.956	—	
Neutrophil-lymphocyte ratio	4.4 (2.3–47.0)	5.3 (0.9–169.7)	0.545	—	
GRWR	1.0 (0.8–2.0)	1.1 (0.6–2.0)	0.054	4.14 (0.27–62.90)	0.305
Transplantation factors					
Transplantation period			0.362	—	
Before 2008	8 (38.1)	22 (27.8)			
After 2008 (2008/01/01)	13 (61.9)	57 (72.2)			
Time of Transplantation			0.588	—	
Daytime transplantation	18 (85.7)	71 (89.9)			
Nighttime transplantation	3 (14.3)	8 (10.1)			
Total operation time (hours)	10.9 (5.1–18.0)	10.2 (5.6–15.5)	0.681	—	
Operative blood loss (L)	3.0 (0.2–18.7)	2.0 (0.1–19.0)	0.130	—	

HBV, hepatitis B virus; HCV, hepatitis C virus; MELD, Model for End-stage Liver Disease; GRWR, graft-recipient weight ratio; OR, odds ratio; CI, confidence interval.

**Table 3 jcm-08-02095-t003:** Analysis of risk factor for 1-year mortality.

	Univariate Analysis			Multivariate Analysis	
Characteristics	Yes, *n* = 31	No, *n* = 58	*p*-Value	OR (95% CI)	*p*-Value
Donor factors					
Age, median (range)	32 (15–52)	27 (15–53)	0.482	—	
Sex (Male:Female)	20:11	46:12	0.129	—	
Cause of brain death			0.641	—	
CVA	24 (77.4)	41 (70.7)			
Traumatic head injury	6 (19.4)	11 (19.0)			
Others	1 (3.2)	6 (10.3)			
Hospital stay (days)	5 (1–36)	5 (1–18)	0.706	—	
Graft factors					
Graft type			<0.001	1.64 (0.38–7.10)	0.507
Left hemi-liver	23 (74.2)	20 (34.5)			
Right hemi-liver	8 (25.8)	38 (65.5)			
Graft weight (gm)	520 (335–1100)	745 (390–1500)	<0.001	0.99 (0.98–1.00)	0.070
Cold ischemia time (min)	513 (102–763)	271 (69–779)	0.039	1.00 (1.00–1.01)	0.069
Warm ischemia time (min)	40 (31–58)	45 (27–65)	0.262	—	
Recipient factors					
Age, median (range)	50 (39–65)	51 (33–65)	0.368	—	
Sex (Male:Female)	18:13	39:19	0.390	—	
Main indication of transplantation			0.173	—	
Alcoholic Liver cirrhosis	6 (19.4)	14 (24.1)			
Virus-related liver cirrhosis	11 (35.5)	18 (31.0)			
Hepatocellular carcinoma	6 (19.4)	20 (34.5)			
Others	8 (25.8)	6 (10.3)			
Viral hepatitis			0.409	—	
HBV	16 (51.6)	25 (43.1)			
HCV	4 (12.9)	14 (24.1)			
HBV + HCV	0	2 (3.4)			
None	11 (35.5)	17 (29.3)			
Child classification			0.885	—	
A, B	15 (48.4)	29 (50.0)			
C	16 (51.6)	29 (50.0)			
Chronic kidney disease staging			0.312	—	
Grade 1, 2	21 (67.7)	45 (77.6)			
Grade 3, 4, 5	10 (32.3)	13 (22.4)			
Hemodialysis before transplantation	3 (9.7)	3 (5.2)	0.419	—	
MELD score, median (range)	18 (7–42)	19 (7–45)	0.887	—	
Biochemical factors					
Sodium (Na, ng/mL)	137 (121–151)	138 (124–152)	0.455	—	
Total Bilirubin (ng/mL)	4.1 (0.5–34.3)	4.6 (0.5–38.2)	0.925	—	
Neutrophil-lymphocyte ratio	4.4 (1.6–169.7)	5.4 (0.9–46.0)	0.535	—	
GRWR	1.0 (0.7–2.0)	1.1 (0.6–2.0)	0.007	3.33 (0.21–51.53)	0.388
Transplantation factors					
Transplantation period			0.832	—	
Before 2008	10 (32.3)	20 (34.5)			
After 2008 (2008/01/01)	21 (67.7)	38 (65.5)			
Time of Transplantation			0.574	—	
Daytime transplantation	28 (90.3)	50 (86.2)			
Nighttime transplantation	3 (9.7)	8 (13.8)			
Total operation time (hours)	10.8 (5.1–18.0)	10.3 (6.4–15.5)	0.856	—	
Operative blood loss (L)	3.0 (0.2–18.7)	2.0 (0.1–19.0)	0.210	—	

HBV, hepatitis B virus; HCV, hepatitis C virus; MELD, Model for End-stage Liver Disease; GRWR, graft-recipient weight ratio; OR, odds ratio; CI, confidence interval

**Table 4 jcm-08-02095-t004:** Recipients transplanted with a hemi-liver graft weighing <580 g and who encountered major events leading to early mortality.

Recipient No.	Age/sex	Indication of Transplantation	MELD	Hemi-liver graft	Graft weight (g)/GRWR (%)	Major events	Mortality (Days)
85	48/F	Autoimmune hepatitis	38	Left	460/0.79	Graft dysfunction, infection	61
111	49/M	HBV, ESLD	21	Left	425/0.85	Graft dysfunction, infection	26
126	39/M	HBV, ESLD	18	Left	520/0.84	Acute renal failure	9
209	65/F	HCV, ESLD	11	Left	410/0.85	Graft dysfunction, infection	13
221	62/F	Wilson‘s disease	17	Left	520/0.91	Graft dysfunction, infection	7
254	50/M	HBV, HCC	9	Left	525/1.17	Graft dysfunction, infection	32
342	54/M	HBV, HCC	11	Left	480/0.81	Antibody mediated Rejection	18
399	48/M	Alcoholic liver cirrhosis	16	Left	335/0.76	Pneumonia	64
426	49/M	HBV, HCC	17	Right	550/0.95	Antibody mediated rejection	9
537	47/M	HBV, ESLD	42	Right	570/0.97	Graft dysfunction, infection	11
538	45/F	PBC	17	Left	450/0.90	Graft dysfunction, infection	31
790	43/F	Alcoholic liver cirrhosis	27	Left	430/0.96	Acute graft versus host disease	46
848	60/F	HCV, HCC	7	Left	520/0.81	HCV relapse	81
978	61/F	HCV, HCC	28	Left	390/0.86	Intra-operative massive bleeding	1
1033	54/F	PBC	29	Left	520/1.18	Intracranial hemorrhage	68

MELD, Model for End-stage Liver Disease; GRWR, graft recipient weight ratio; F, female; M, male; HBV, hepatitis B virus; HCV, hepatitis C virus; ESLD, end-stage liver disease; PBC, primary biliary cirrhosis; HCC, hepatocellular carcinoma.

**Table 5 jcm-08-02095-t005:** Clinical features of patients on the wait-list and liver transplantation.

	Waitlist	SPLT	WLT	
	*n* = 995(%)	*n* = 100(%)	*n* = 165(%)	*p*-Value
Age (years), median (range)	53 (25–66)	50 (33–65)	53 (19–65)	0.285
Male: female	750:245	63:37	132:33	0.006
Hepatitis status				0.061
HBV	476 (47.8)	45 (45.0)	93 (56.3)	
HCV	167 (16.8)	19 (19.0)	30 (18.2)	
HBV + HCV	35 (3.5)	2 (2.0)	9 (5.5)	
None	317 (31.9)	34 (34.0)	33 (20)	
Main indications for LT				0.116
Acute hepatic failure	48 (4.8)	1 (1.0)	2 (1.2)	
Hepatocellular carcinoma	261 (26.2)	29 (29.0)	55 (33.4)	
Viral cirrhosis	379 (38.1)	30 (30.0)	81 (49.1)	
Alcoholic cirrhosis	172 (17.3)	24 (24.0)	20 (12.1)	
Others	135 (13.6)	16 (16.0)	7 (4.2)	
MELD score, median (range)	14 (3–42)	19 (7–45)	22 (7–54)	<0.0001
Medical priority				0.289
Status 1	4 (0.4)	0 (0.0)	2 (1.2)	
Non-status 1	991 (99.6)	100 (100.0)	163 (98.8)	
1-year mortality	593 (59.6)	31 (31.0)	43 (26.0)	<0.0001

SPLT, split liver transplantation; WLT, whole liver transplantation; HBV, hepatitis B virus; HCV, hepatitis C virus; LT, liver transplantation; MELD, Model for End-stage Liver Disease.

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
