# Peer review of "Encouraging Split Liver Transplantation for Two Adult Recipients to Mitigate the High Incidence of Wait-List Mortality in the Setting of Extreme Shortage of Deceased Donors"

_jcm, 2019, doi:10.3390/jcm8122095_

Round 1

Reviewer 1 Report

I read with great interest the manuscript by Kun Ming Chan et al. This is a very well written manuscript , has a sound merit, is clinically relevant and has wide applications to resource utilization and organ allocation in the current climate of organ shortage. 

The content is applicable and will be of interest to the global audience.

There are some concerns worth addressing: 

1) the 1 year mortality rate is extremely high pos-LT. can the authors explain? Is this outcome (21 and 31%) early and 1 year mortality a norm? What is the overall mortality in Taiwan/institution with a whole liver graft? 

2) there appeared no difference in donor criteria and recipient level of sickness.  The only thing that was evident is the weight of the graft made a dramatic improvement in survival. was there any difference in the cardiovascular status of those who survived and those who died? Is it routine to do ECHOs or caths in the recipients?

3) it looks like a smaller graft lead to poor outcome. The aetiology is" graft dysfunction". can that be characterized further? was the graft failure due to: vascular thrombosis? 

Reviewer 3 Report

Materials and Methods:

The number of patients investigated is Part of the results

how did you choose the recipients of SPLT.

Did the recipients consent to this potentially more dangerous procedure?

what kind of donors were used? How was the selection process for donation in general?

would it be possible to provide a illustration of the splitting procedure?

How was early mortality defined?

Please describe multivariate analysis protocol. Maybe consider using a more complex stepwise procedure. Collets model for selection?

Results:

the application of the multivariate analysis is not clear to the reader. Please explain the results especially with regard to the methodology used why is p<0.1 considered significant? 

Please compare your results to standard postmortal transplant.

deaths on the waiting list seem to be very high. Please explain.

please provide a comparison between patients on the waiting-list and those transplanted with SPLT.

Discussion

„Advances in the surgical instruments used for liver resection undoubtedly play a crucial role in hepatobiliary surgery as well as in the utilization of partial liver grafts for LT. Specifically, the standard technique of liver resection nowadays involves the use of an ultrasonic dissector that enables identifying the major vascular and biliary structures during the transection of the hepatic parenchyma, thus reducing the surgical complications. Moreover, the accumulated experience and considerable volume of major liver resection and LDLT procedures had encouraged the use of SPLT at our institute.(30, 31) Therefore, splitting one liver into full right and full left hemi-liver grafts is now performed whenever possible, taking into consideration the high number of adult patients awaiting for LT and the very small number of deceased donors in the transplantation center.“ consider revising shortening or cutting from the manuscript

please truly discuss the limitations of your work.

consider revising the summary/conclusion to be more consize.
